# A 2D-Lidar-Equipped Unmanned Robot-Based Approach for Indoor Human Activity Detection

**DOI:** 10.3390/s23052534

**Published:** 2023-02-24

**Authors:** Mondher Bouazizi, Alejandro Lorite Mora, Tomoaki Ohtsuki

**Affiliations:** 1Faculty of Science and Technology, Keio University, Yokohama 223-8522, Japan; 2Graduate School of Science and Technology, Keio University, Yokohama 223-8522, Japan

**Keywords:** activity detection, healthcare, 2D Lidar, deep learning, machine learning

## Abstract

Monitoring the activities of elderly people living alone is of great importance since it allows for the detection of when hazardous events such as falling occur. In this context, the use of 2D light detection and ranging (LIDAR) has been explored, among others, as a way to identify such events. Typically, a 2D LIDAR is placed near the ground and collects measurements continuously, and a computational device classifies these measurements. However, in a realistic environment with home furniture, it is hard for such a device to operate as it requires a direct line of sight (LOS) with its target. Furniture will block the infrared (IR) rays from reaching the monitored person thus limiting the effectiveness of such sensors. Nonetheless, due to their fixed location, if a fall is not detected when it happens, it cannot be detected afterwards. In this context, cleaning robots present a much better alternative given their autonomy. In this paper, we propose to use a 2D LIDAR mounted on top of a cleaning robot. Through continuous movement, the robot is able to collect distance information continuously. Despite having the same drawback, by roaming in the room, the robot can identify if a person is laying on the ground after falling, even after a certain period from the fall event. To achieve such a goal, the measurements captured by the moving LIDAR are transformed, interpolated, and compared to a reference state of the surroundings. A convolutional long short-term memory (LSTM) neural network is trained to classify the processed measurements and identify if a fall event occurs or has occurred. Through simulations, we show that such a system can achieve an accuracy equal to 81.2% in fall detection and 99% in the detection of lying bodies. Compared to the conventional method, which uses a static LIDAR, the accuracy reaches for the same tasks 69.4% and 88.6%, respectively.

## 1. Introduction

With the population pyramid in most countries of the world shifting towards a constrictive shape, more and more pressure is being put on healthcare staff to monitor elderly people. Elderly people, in particular ones living alone, need continuous monitoring to make sure they are in good condition and not going through hazardous events. Falling, for the most part, is considered to be one of the most dangerous events that would occur. This is because the elderly person might lose consciousness or suffer from other issues, which might be either the root cause of falling (e.g., heart attack, etc.) or a direct consequence (e.g., injuries). A report from the World Health Organization [1] showed that 37.3 million falls that are severe enough to require direct intervention happen every year. Over 684,000 of these are fatal and constitute the second main cause of death due to unintentional injury, second only to road traffic injuries. Even though the remainder of the falls are not lethal for the most part, they contribute to over 38 million disability-adjusted life years (DALYs). Detecting such an event is, therefore, of great importance to avoid long-term lethal consequences.

With that in mind, the detection of falls, and more generally the activities of elderly people living alone, has been a hot topic of research. In this context, the expression “activity detection” has been conventionally used to refer to the detection of one’s actions, intentionally or unintentionally performed [2,3]. The level of precision of the activities being detected varies widely. A coarse-grained classification aims to classify the activity “fall” from any other activity, whereas a fine-grained one aims to identify the activity at a very fine level, such as whether the person is walking, standing, sitting, etc. Several approaches have been proposed in the literature to perform activity detection [4,5,6,7,8,9]. Such approaches usually rely on sensing devices placed in the environment of the monitored person or attached to their body. For instance, with the development of smart wearable devices, multiple approaches make use of the gyroscopes and accelerometers installed in devices such as smartphones and smartwatches to perform classification tasks aiming to identify whether or not a person falls/is falling [10,11,12]. However, their requirement to be maintained and well placed all the time presents a burden for the monitored person and might not be very reliable. This makes approaches that rely on sensors placed in the environment and attached to a continuous source of electrical power a more attractive option. Over the years, approaches relying on cameras, sensors, and even ambient signals have been proposed [2,13,14,15]. Such approaches have their limitations as we will explain later in Section 2.1. However, these can be summarized into the following: (i) coverage problems, (ii) privacy problems, (iii) practicality problems, and (iv) cost problems.

On a different topic, robotics are considered to be some of the trendiest, most popular and fastest growing technologies today [16]. Thanks to their capabilities, mobile robots can replace or operate alongside humans in numerous environments. Mobile robots can move independently in an industrial facility, laboratory, house, etc., that is, without aid from external human operators. In recent years, such robots have seen a raise in usage in houses, mostly for cleaning purposes. Roaming in the house and cleaning the floor has been the objective of many such robots that have been a commercial success over the last few years [17]. However, the applications of such robots (i.e., cleaning robots) could be extended to tasks beyond their basic ones, namely the topic of this paper’s research: human activity recognition (HAR). These robots can be equipped with sensors and computation devices that could be tuned to identify the activities of people in indoor environments. This is thanks to the advances in both sensing technologies and artificial intelligence (AI) [18,19,20].

In this context, the Robot Operating System (ROS) [21] refers to a set of open-source software libraries and tools that simplify the process of building robot applications, allowing for the integration of new components into the robot without interfering with its basic functioning. This work is dependent on the ROS to provide the basic functionalities of mobile robots during our simulations. In addition, physics simulations are fundamental for robotics research and industry alike and constitute the second main component of our simulations. Physics simulators allow the reproduction of real-world environments since they give an environment that is affordable and provides users access to several desired robots without the problem of deteriorating or harming the physical platform.

Our current paper falls in the intersection between these key research areas: HAR and robotics simulations. We introduce our in-house-built simulator to reproduce the behavior of autonomous robots and the functioning of the sensors used for our work. Our simulator makes use of a realistic physics engine [22,23,24] to recreate a real-world-like environments in which subjects and robots can move and act as they would in the real world. Our simulator is then used to simulate several scenarios in which a person is performing several activities in a room while a cleaning robot equipped with a 2D light detection and ranging (LIDAR) cleans the room. The robot is simulated to be equipped with a computation device, which uses the information collected by the 2D LIDAR (hereafter referred to as “2D Lidar” or simply “Lidar”) to classify, in real time, the activities performed by the person.

The novelties of this paper can be summarized as follows:We introduce a novel approach for indoor HAR using an unmanned robot equipped with a 2D Lidar. Unlike conventional methods [4,8], which rely on static 2D Lidars placed on the ground, we rely on a moving robot (such as a cleaning robot) to achieve this goal, making the approach more practical.We address the different challenges that come along with such an approach, namely (i) the continuous movement of the Lidar, which makes it hard to keep track of the location of the subject, (ii) the cases when the fall activity might occur when the falling person is hidden by obstacles, and (iii) the fact that the manifestation of an activity from the Lidar’s perspective varies greatly depending on the relative position between the subject and the Lidar.Through simulations, we demonstrate the effectiveness of our proposed approach by comparing it to the conventional approaches [4,8].We evaluate different tuning parameters for our proposed approach and derive the optimal configuration for better activity detection.We introduce a novel simulator that simulates the behavior of cleaning robots equipped with sensors in the presence of a humans in indoor environments.

The remainder of this paper is structured in the following order. In Section 2, we describe some of the state-of-the-art work related to the current task that we are researching and introduce our motivations for this work. In Section 3, we introduce the key concepts allowing for a better understanding of our proposed approach. In Section 4, we present how our simulator is built. In Section 5 and Section 6, we describe in detail our proposed method for localization and activity detection and evaluate it using our simulator, respectively. Finally, in Section 7, we conclude this paper and present directions for a future work.

## 2. Related Work and Motivations

### 2.1. Related Work

A key concept in HAR is the detection of existing people in the first place and the identification of their location. Many approaches for localizing a person from a mobile robot using one or multiple sensors have been studied over the past years [25].

Most of these approaches rely heavily on privacy-threatening devices, namely common RGB cameras and depth cameras, to recognize the humans, locate them, and identify their activities. To address the privacy issues of RGB and RGB-D cameras, we aim to perform two of these tasks, namely localization and activity recognition using non-privacy-invasive devices. In particular, a 2D Lidar has been used in a previous work of ours [4] to localize people and recognize their activities. In that paper [4], we proposed a novel approach that uses 2D Lidar and DL to perform activity recognition, namely fall detection. The 2D Lidar is placed on the ground so that some feet points are seen. Throughout our experiments, we demonstrated that it is possible to accurately identify up three different people, to detect unsteady gait (i.e., when the person is about to fall or feeling dizzy), and to distinguish between four other activities: walking, standing, sitting, and falling. The three classification tasks have reset al.ults of over 90% accuracy. One of the limitations in that work is the small amount of data used, which does not promise the generalizability of the proposed approach. Another limitation is the environments used. Only five different empty rooms with a few obstacles were used, in which the 2D Lidar was static and only a single person was present at a time.

In a similar way, Luo et al. [8] proposed a Lidar-based approach that recognizes the activities performed by multiple people simultaneously by classifying their trajectories. In their work, the authors implemented a Kalman filter and built two neural networks. The proposed temporal convolutional network (TCN) achieved the best result of 99.49% in overall accuracy. Their work, however, defines “activity” quite differently from what is conventionally agreed on. Rather than identifying the nature of the action itself, they identify movements from one point in the room to another or standing in one point for a certain period. This does not only exclude the objective defined here (i.e., identifying the action performed) but it also means that the approach cannot generalize unless retrained in every new encountered environment.

Okusako et al. [26] proposed an approach that tracks humans walking around a mobile robot using a 2D Lidar in real time by converting the data captured by the Lidar in polar coordinates (r,θ) to a 2D image with (x,y) coordinates. Then, human tracking is achieved via block matching between templates, i.e., appearances of human legs, and the input polar coordinates data. A particle filter is also employed to increase the robustness in case of occlusions. This approach, however, requires the human to be within a limited range from the 2D Lidar to be accurately detected. In addition, similar to [4], this approach is not capable of detecting the human when it is out of its line of sight (LoS). Finally, this approach is not meant to detect the activity of the human, but solely to locate them, leaving the task of activity detection a task to be performed by other means.

The approaches by Arras et al. [27] and by Taipalus et al. [28] define a person as a model composed of two legs and use Kalman filters, constant velocity motion models, and a multi-hypothesis data association strategy to track the person using a 2D Lidar as well.

Finally, one of the most widely known methods for detecting legs on mobile robots is based on [7] and is implemented as a ROS package with the name of “Leg Detector” (https://github.com/marcobecerrap/edge_leg_detector (accessed on 1 February 2023)). Although the original paper implements multi-sensor data fusion techniques, which combine an onboard laser range finder (LRF) to detect the legs and a camera to detect the face, the ROS package only uses the 2D Lidar as the input to detect the legs. A newer approach by Leigh et al. [29] uses a different algorithm that uses only a 2D Lidar. The algorithm is validated in varied surroundings, both indoor and outdoor, and on different robot platforms: the intelligent power wheelchair and a Clearpath Husky. The method is also released as a ROS package (https://github.com/angusleigh/leg_tracker (accessed on 1 February 2023)). Finally, this algorithm was improved again in [30] to become named PeTra. This time, an offline trained full CNN is used to track the pairs of legs in cluttered environments. This method is also published as a ROS package [31] and has become the default approach for detecting legs using a 2D Lidar from a mobile robot. These, however, do not address the task of activity detection in the way we do in this current work.

### 2.2. Motivations

Given the limitations of the works described in Section 2.1, we aim to tackle the task of activity recognition, in its broader meaning, in a realistic room environment where a person performs all sorts of activities using the Lidar technology. Other works [4,8] require the Lidar to be placed in a well-chosen single location, which allows it to capture the person’s data points all the time. This is not practical, and there is usually no guarantee that such a place exists in the first place, in particular where a variety of furniture exists. Nonetheless, these approaches require a dedicated sensor to be used. This leads us to believe that the use of an already-existing device in the environment could be a much more practical solution. For instance, cleaning robots are getting more and more commonly used in houses and their cost has been decreasing over the past years. Being able to use them to perform HAR would make them a great alternative for the usage of dedicated sensors. Nevertheless, having the ability to roam in the house allows the robot to detect dangerous activities, such as falling, even in remote places, which might be beyond the coverage areas of conventional sensors. Another merit for using sensors on a moving robot is the ability to detect fall activities even after they occur. Unlike conventional techniques that require the detection when the event happens by relying on changes in the environment during the event of falling, it is possible for the robot-mounted 2D Lidar to detect the rough shape of a person lying on the ground and recognize that a fall has occurred even several seconds/minutes later.

With that in mind, our objective is to use mobile robots to exploit their capability of roaming freely in indoor environments to perform HAR, namely the detection of fall events. For this sake, we built a simulator that allowed us to recreate realistic indoor environments and simulate the behavior of a 2D Lidar mounted on top of a robot. We used our built simulator to create a multitude of scenarios in which a person performs all sort of activities, which we identified using a DL-based classifier. For the data to be classified by the DL-based classifier, they were cleaned up via a clustering algorithm, interpolated using a customized algorithm to make them independent of the robot’s location, and divided into small segments, each to be classified based on the activity performed. These steps will be clarified in much detail in Section 5.

In the following section, we introduce some of the key concepts related to this work. We clarify how 2D Lidars and mobile robots operate and explain the principle of the ROS. Afterwards, we proceed to introduce our simulator, our proposed method for HAR, and the simulation results obtained.

## 3. Key Concepts

### 3.1. Lidar Technology

A Lidar is a device that measures the distance to the nearest obstacles by emitting light and measuring the time it takes to reflect from objects and reach the receiver.

Knowing the location and orientation of the Lidar itself in an absolute referenceand the coordinate of the point of reflection can be derived. A 2D Lidar is simply a Lidar mounted on a rotating device, allowing it to perform the emission/reception and computation procedure numerous times, thus building up a 2D map (typically along a horizontal plane) made up of all the points that the Lidar captured. Depending on the rotation speed, the refresh frequency of the detection can range from 1 Hz (i.e., for 1 rotation per second) to 100 Hz (i.e., for 100 rotations per second). In a similar way, the angular resolution of the Lidar reflects how precise the 2D Lidar can be on its rotating range. Regardless of the required precision, farther objects are always represented by fewer points compared to closer objects. Moreover, objects that are behind another object from the Lidar are not hit by the ray so they are not seen by the Lidar.

In Figure 1, a simplified pictorial representation of the functioning of the 2D Lidar is shown. The position of an object’s point *p* in Cartesian coordinates $x_{p}$ and $y_{p}$ with reference to the 2D Lidar position and orientation can be obtained using the following formula:(1)$$\begin{array}{ccc} x_{p} & = & r_{p}\;\cdot \;\cos\left(\theta_{p}\right),\\ y_{p} & = & r_{p}\;\cdot \;\sin\left(\theta_{p}\right), \end{array}$$
where $r_{p}$ and $\theta_{p}$ are the distance and the angle from the 2D Lidar to the point *p*, respectively.

Typical Lidars available in the market have a range of detection from 0.2 to 25 m, with an error of between 1% and 2% depending on the measured distance (that is, the higher the distance is, the higher the error is).

### 3.2. Mobile Robots

In our work, we assume that our 2D Lidar is mounted on an autonomous mobile robot. In practice, such a robot can be a cleaning robot (e.g., a robot vacuum cleaner). An autonomous mobile robot is capable of perceiving its environment through its sensors, processing that information using its onboard computer, and responding to it through movement.

A simple differential drive robot with two fixed standard individually motorized wheels and a castor wheel is used for the sake of our work. To specify the position of the robot on the plane, we establish a relationship between the global reference frame of the plane and the local reference frame of the robot, as shown in Figure 2. The way the robot keeps track of its location in the global reference is not discussed here. However, the relationship is assumed to be modeled mathematically and is part of the simulator library. The robot internals and mechanics are not discussed here either. However, further details about autonomous mobile robots can be found in [32,33].

While out of the scope of the current paper as we will use pre-built packages for controlling the robot, it is important to mention that the robot behavior and motion are not targeted towards detecting the human’s location or activity, but rather towards performing its original task, namely cleaning the house. Therefore, it is safe to assume that the robot roams arbitrarily, though its path should cover the same place multiple times as little as possible.

### 3.3. Robot Operating Systems (ROS) and Unity ROS Simulation

Mobile robots are complex systems that combine motors, sensors, software, and batteries, which must all work together seamlessly to perform a task. A set of open-source software libraries and tools that simplify the process of building robot applications is the ROS [21]. The ROS supports hardware interfaces for many common robot components, such as cameras, Lidars, and motor controllers. This allows us to focus on addressing specific problems and implementations by not reinventing the wheel for the rest of the modules. The ROS 2 is an updated version of the ROS containing more features and components allowing for covering more of the functionalities of mobile robots [23].

The Unity ROS Simulator [34] is used in this work to generate synthetic data. Starting from August 2021, Unity added official support for the Robot Operating System 2 (ROS 2) [23,34] on top of many already existing plugins and solutions for simulating real-life environments. This allows the use of mobile robots packages from the ROS that include communication modules, navigation, simultaneous localization and mapping (SLAM), and multiple sensor implementations and visualization such as 2D and 3D Lidars. Even if some robotics characteristics are not as precise as in other simulators, the requirement of creating hundreds of variations of humans performing everyday life activities in randomized scenarios is crucial for this application, which cannot be achievable by using other simulators.

In this work, the Unity Perception package is used to simplify and fasten the process of generating synthetic data sets for computer vision applications by delivering an easy-to-use and highly configurable toolbox. This open-source application enhances the Unity Editor and engine components to provide correctly annotated examples for numerous typical computer vision tasks.

## 4. Simulator

In this section, we will describe the process of building the customized simulator that we used in the rest of this paper.

The first step in this work is to create a simulator program that generates realistic 2D Lidar data from a mobile robot while a person is performing different activities in an automated form. By automated, we mean that the simulator can run one scenario after another while communicating with the ROS 2 and saving the data from the different scenarios automatically.

Needless to say, in the field of robotics, several simulators have been very successful and were used in several works to simulate autonomous robots. Gazebo [35], in particular, has been attracting most of the attention given its rich libraries and the community behind it supporting its features and iterating over them. Other simulators, such as MuJoCo [36] and CARLA [37], received less attention, though they have their respective applications. Our early experiments that led to this work were conducted using Gazebo and were later on discarded. This is because our work does not only require a faithful simulation of the robot behavior and mechanics, but also a similarly faithful simulation of the human behavior and motion, which was not possible given the focus of Gazebo in robotics. Nevertheless, to create a decently sized data set and automate the process of such a creation, while satisfying a certain degree of randomness in the scenarios created, Gazebo, despite being usable, is not flexible enough to allow the automatic implementation of such scenarios.

That being the case, we opted for using a simulator based on top of a realistic physics engine that reproduces the human motion and activities faithfully, while supporting robotics simulations.

Throughout our work, the Unity game engine and its recently added Robotics Simulation package with the Robot Operating System 2 (ROS 2) [38] were the most suitable for creating more complex, realistic, and useful scenarios.

### 4.1. Activity-Related Data Collection

The first step towards building the simulations is to create a set of 3D animations representing the activities using a 3D modeling tool that can be imported and used. In this context, the Archive of Motion capture As Surface Shapes (AMASS) database of human motion [39] is used. The data are input to Unity. The animations of the human bodies are manually labeled specifying for each frame the nature of the activity being performed. By connecting activities to one another seamlessly, Unity can recreate a real human set of activities with no sudden changes in the positions of the body parts.

### 4.2. Physics Engine

As stated above, throughout this work, we use Unity to run our scenarios and simulations. While Unity is basically a game engine made to allow creators to create games scripted in C#, it has other usages, including building simulators for robotics, autonomous cars, etc. [23,24]. In this context, our simulators falls in the first category (i.e., robotics) while exploiting more features in Unity, namely the physics engine PhysX and its two features raycast and raytracing. These latter two features allow us to reproduce, faithfully, the behavior of a Lidar. At any given moment, it is possible to cast a ray originating from the Lidar’s supposed emitter and have realistic reflections, which are then captured by the receiver and used to measure the distance to the surroundings in a given direction. This simulates the behavior of the Lidar more perfectly than in real-world scenarios as our experiments in [4] have shown. In real-world scenarios, noise and multiple reflections lead to erroneous measurements and the detection of points that do not actually exist. To account for this realistic behavior of light, we introduce several parameters to our simulator to have similar patterns. These parameters are the following:The distance measurement error $d_{\mathrm{error}}$, referring to the inaccuracy in the estimation of the distance and presented in the percentage of the actual distance measured by the simulation Lidar.Missing data points $N_{\mathrm{lost}}$, referring to the number of points for which the Lidar emits a light beam but does not receive the reflection, presented in the percentage of data points that are lost due to bad reflection.Wrong data points $N_{\mathrm{extra}}$, referring to the number of points where the laser beam was reflected on multiple objects and was received in the wrong angle, leading to a wrongly detected point.Angle inaccuracy $\theta_{error}$, referring to how different the measured angle is from the actual one. This is because the Lidar keeps on rotating, and while we assume the exact same angles for each rotation, this is not the case in the real world. It is also presented in an interval of error $\left[-\frac{\theta_{error}}{2},\frac{\theta_{error}}{2}\right]$.

### 4.3. Robot Simulation

Regarding the robot, the TurtleBot3 Pi mobile robot [40] is used because it is mechanically similar to a commercial vacuum robot while being open-source and ROS-compatible. It is widely used in the research community in robotics areas, such as autonomous navigation, manufacturing monitoring, healthcare, education, etc. Plugins for these sensors have been developed by the active ROS community and the corresponding simulators for the simulation environments and the real sensors were integerated. In order to use this robot in the Unity simulator, an imported Unified Robot Description Format (URDF) file is loaded to Unity using the URDF-Importer package from Unity. Then, the user must modify the generated model in case of problematic mesh issues and add custom sensors’ scripts. The localization algorithm proposed in the next section requires a map of the room where the simulations take place. This map can be retrieved from the ROS 2 Nav2 package by simply calling a ROS 2 Service.

### 4.4. Additional Features

A couple of additional features have been implemented in the simulations. First, a low-resolution video is recorded from a corner of the room so that it is easy to then manually check the simulations in case some data are not correct. Second, the ground-truth pose of the human (position and orientation) is saved on a file to later compare it to the localization algorithm. Third, two more 2D Lidars have been added to the robot at different heights so that their data can be compared in the future. They are located at ground level at a height equal to 0 cm and at a height equal to 25 cm, respectively. For reference, the main Lidar used in our work is placed at a height equal to 10 cm.

### 4.5. Simulator Output

#### 4.5.1. Naming Convention

For a good understanding of the simulator output, we use the following terminology that was adopted in previous works [4]:Measurement point:A measurement point *p* is the individual point measured at a time *t* by the Lidar. It is represented as the tuple p=(r,θ), where *r* is the measured distance and θ is the angle of measurement. This measurement point is measured from the Lidar’s perspective.Scan: A scan is the set of measurement points within a single rotation of the Lidar. Given the high speed of rotation $\rho_{s}$ (i.e., over 10 Hz), we could assume that all measurement points taken during one rotation correspond to a single point in time *t*. A scan S(t) can therefore be represented as the tuple
(2)$$S\left(t\right)=\{p_{i}=\left(r_{i},\theta_{i}\right),\;\;\;\;\;\;\;\;i=\left[1,\cdots ,N\right]\},$$
where $p_{i}$ is the ith measurement point and *N* is the total number of measurement points during the rotation. Theoretically, *N* is dependent on the rotation, as the number of measurement points collected in one rotation is not always the same; therefore, *N* should rather be referred to as N(t). However, in our work we do normalize the scans via interpolation to have exactly N=720 so that, for each 0.5${}^{\circ }$, a measurement point is recorded (i.e., $\theta_{i}\in \left\{0^{\circ },0.5^{\circ },1^{\circ },\cdots ,359.5^{\circ }\right\}$).

#### 4.5.2. Output Data

With that in mind, we could proceed with defining the simulator’s output. In summary, for each scenario, the simulator outputs two sets of information that we use either for running our approach or as a ground truth to evaluate it.

The information generated that is used by our algorithm is the following:Lidar-generated data points: They include the scans that the Lidar mounted on the robot generate. This is a set of scans $S=\{S\left(t\right),\;\;\;\;\;\;t=\left[0,\cdots ,N_{t}\right]\}$, where $N_{t}$ is the number of rotations performed by the Lidar during the entire duration $T_{scenario}$ of the scenario. $N_{t}=T_{scenario}\times \rho_{s}$.Lidar location during the scenario: This is interpolated in a way where a pair of coordinates $\left(x_{l}\left(t\right),y_{l}\left(t\right)\right)$ is generated every time a scan is generated; that is, the location of the Lidar is reported alongside with each individual scan. This is a set $L_{l}=\left\{\left(x_{l}\left(t\right),y_{l}\left(t\right)\right),\;\;\;\;\;\;t=\left[0,\cdots ,N_{t}\right]\right\}$. The Lidar location is supposed to be known at any given moment, as the robot is expected to be self-aware of its position within the house/room for cleaning purposes.

The information generated that is used for evaluation is the following:Subject’s location: Similar to the way we generate the coordinates of the Lidar, for every time step *t* (the time of the scan collected by the robot), the location of the subject is reported. This is also a set of coordinates $L_{s}^{gt}=\left\{\left(x_{s}^{gt}\left(t\right),y_{s}^{gt}\left(t\right)\right),\;\;\;\;\;\;t=\left[0,\cdots ,N_{t}\right]\right\}$.The activities performed by the subject: During the annotation, each activity happens over a certain duration of time, and all scans that took place during that duration will be given the label associated with that activity. This translates into the set $A^{gt}=a^{gt}\left(t\right),\;\;\;\;\;\;t=\left[0,\cdots ,N_{t}\right]\}$, where $a^{gt}\left(t\right)$ is the ground-truth activity performed during the time step *t* of the scenario.Room map: This is a simple matrix-shaped image, with a certain degree of precision indicating whether or not there is an obstacle in a given position. Each pixel contains a value set to 1 or 0, indicating whether or not there is an obstacle in the location of the room corresponding to that pixel. A map of shape $X_{px}\times Y_{px}$ pixels for a room of dimensions $d_{length}\times d_{with}$ indicates that each pixel represents a rectangle of the room of the size $\frac{d_{length}}{X_{px}}\times \frac{d_{width}}{Y_{px}}$.

## 5. Proposed Method for Human Localization and Activity Detection

In this section, we introduce our methods for human localization and activity detection.

### 5.1. Localization

The first step in the proposed method is to localize the human regardless of the activity being performed, even behind objects. The localization of the subject is a very important task on its own. It also allows for a reliable classification task later on. Our algorithm consists basically of two steps: (i) knowing the Lidar’s coordinates and orientation, we locate the data points collected from the surroundings relative to the Lidar and transform them to absolute ones in a reference attached to the room and (ii) use previous knowledge and/or a clustering algorithm to locate the user-related data point. The input data to this algorithm is what has been described in the previous section (i.e., the Lidar-generated data points and its location during the scenario) as well as any previous knowledge acquired from previous scans.

A flowchart of the proposed algorithm for the location identification is given in Figure 3. We first transform the coordinates of the data points in a given scan *S* collected by the Lidar into absolute coordinates. This is straightforward knowing the location of the Lidar itself. Given the location of the Lidar $\left(x_{l}\left(t\right),y_{l}\left(t\right)\right)$ and a data point $p_{j}$ in the scan whose polar coordinates are $\left(r_{j}\left(t\right),\theta_{j}\left(t\right)\right)$, the Cartesian coordinates of the data point *j* in the absolute reference are calculated as follows:(3)$$\begin{array}{ccc} x_{j}\left(t\right) & = & x_{l}\left(t\right)+r_{j}\left(t\right)\cos\left(\theta_{j}\left(t\right)\right),\\ y_{j}\left(t\right) & = & y_{l}\left(t\right)+r_{j}\left(t\right)\sin\left(\theta_{j}\left(t\right)\right). \end{array}$$

By doing this, we are able to transform the scan *S* from polar coordinates with reference to the Lidar to a set of points with their absolute Cartesian coordinates. For simplicity, we refer to this new set as a scan as well.

Afterwards, we use a slight alteration of the algorithm we previously proposed in [4] to clean the scan, discard noisy data points, identify the subject’s related data points, and derive their location. Algorithm 1 shows how this is peformed. The algorithm first compares the data points of the current scan to an empty scan taken when no person was in the room. The data points that are not within a range of $\delta_{d}$ to the ones of the empty scan are considered as newly found points and are saved in a separate list $S^{*}$. The clustering algorithm DBSCAN [41] is used to find clusters of newly found points. Depending on the number of clusters and the potentially found location in the previous scans, the subject’s location is retrieved and is used to isolate the subject’s data points in the current scan. 

**Algorithm 1:** Location identification

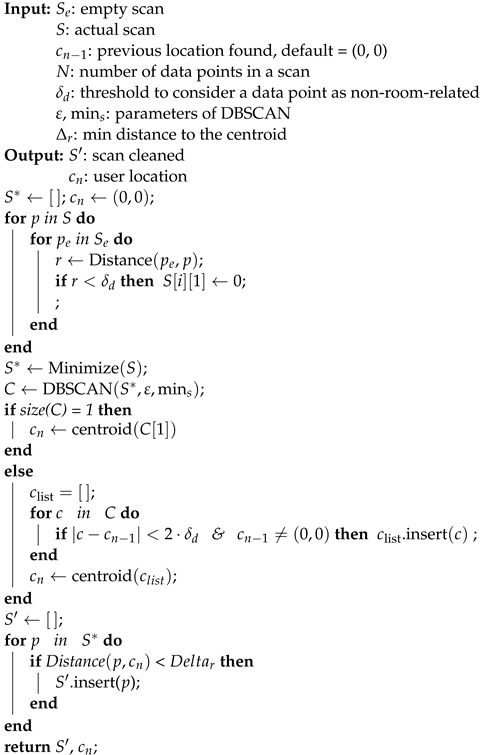



### 5.2. Activity Recognition

The second step in the proposed method in this work is to recognize the activities performed from the person’s points identified by the Lidar placed on top of a mobile robot (i.e., the output of the previous step). Identifying the activity performed by the subject requires the localization algorithm to perform well. The quality of the output generated by the algorithm affects largely the performance of the activity identification. Our approach uses deep learning techniques for classifying the following human activities: crawling, falling down, getting up, lying down, running, sitting down, standing, walking, and walking unsteadily. Only one type of network is used: a long short-term memory (LSTM) neural network architecture. This type of network deals with the data after pre-processing and interpolation. LSTM networks are basically an improved version of conventional recurrent neural networks (RNN), made to learn and keep in memory long sequences of input data and learn the dependencies between them. In this work, three different variations of a similar LSTM model are proposed by using three different inputs. The architecture of the models is very simple, containing only a few layers. Although infinite input data can be generated from the simulations, the animations used for some activities are scarce. Therefore, a simpler architecture is preferred to avoid overfitting.

Some other works using a static 2D Lidar, such as [4], use the output of the scans in their original polar coordinates for the input. In our case, however, this is not possible as the 2D Lidar is moving all the time and the ranges of the angles change. For instance, in cases where the robot is moving and the subject is standing, the position of the subject from the Lidar’s perspective is changing, which, in [4], is an indication that the subject is moving.

For the three variations of our model, the input is the cleaned-up scan $S^{\prime }$ showing a map containing only the subject’s data points in Cartesian coordinates as returned by the localization algorithm. As the input shape has to be constant, a certain interpolation is used to create a fixed input type/size. Therefore, we apply a simple technique, with 3 variations to transform the set of coordinates representing the data points of the user in a single scan into a 2D image. This, in return, means that a sequence of scans could be transformed into a sequence of 2D images (i.e., an animation) that can be processed by a convolutional LSTM (ConvLSTM) network. The variations of the transformations of the individual scans are conducted as follows:Boolean grid: in this variation, we create a grid of a fixed size supposedly fitting the largest room possible, similar to how we described a map in Section 4.5.2, given a certain required degree of precision, each pixel indicating whether or not there is a data point in the part of the map that corresponds to that pixel. In other words, each pixel contains a value set to 1 or 0, indicating whether or not there is an obstacle in the location of the room corresponding to that pixel. A map of shape $X_{px}\times Y_{px}$ pixels for a room of dimensions $d_{length}\times d_{with}$ indicates that each pixel represents a rectangle of the room of size $\frac{d_{length}}{X_{px}}\times \frac{d_{width}}{Y_{px}}$.Image with Circles: in this variation, we assume a white image with a size of $X_{px}\times Y_{px}$ pixels. Assuming a larger image size in this variation, plotting small dots indicating the user data points make them almost invisible. We therefore plot each data point with a circle whose center is the coordinates of the data points and with a radius $R_{c}$ large enough to make it visible in the image.Image with ellipses: this variation is similar to the previous one. The only difference is that we use ellipses with radii $\left(R_{a},R_{b}\right)$ whose centers are the coordinates of the data points themselves.

For the classification, not only do the images need to have the same size, but the lengths of their sequences themselves need to be the same. Therefore, we use a sliding window with a fixed length to cut the sequences into equal sizes. Later, when we run the evaluation, we will describe in more detail the data acquisition.

For the first variation of the technique, we use a grid of a size equal to 60 × 60 pixels. For a room of 6 × 6 m^2^, this translates into a precision of 10 × 10 cm^2^ per pixel, meaning that every 10 × 10 cm^2^ of the room is represented by a single pixel As for the other two variations, we assume different image sizes, though the reported results will be when using 1200 × 1200 pixels. While this is a technical detail that we will elaborate on later in the next section, we mentioned it to explain why we use a different number of convolutional layers in our ConvLSTM network.

The first variation uses the neural network architecture shown in Figure 4. It consists of a single time-distributed convolution with 32 filters whose size is 3 × 3, followed by a time-distributed 2 × 2 max pooling layer. The output of the max pooling layer is flattened and fed to the LSTM layer, which has 40 cells. The LSTM output is connected to 2 fully connected layers that have, respectively, 100 and $N_{C}$ neurons. $N_{C}$ represents the total number of classes of activities.

The second and third variations use the neural network architecture shown in Figure 5. It consists of 3 blocks of time-distributed convolutions, where each block has 2 time-distributed convolutional layers followed by a single max pooling layer. The number of filters as indicated in the figure for the layers in order is (64, 64, 32, 32, 32, 32). All filters have a size equal to 3 × 3. The time-distributed max pooling layers use a pool of size 2 × 2. The output of the max pooling layer in the third block is flattened and fed to the LSTM layer, which has 40 cells, which is followed by 2 dense layers of the same dimension as the previous ConvLSTM.

## 6. Evaluation of the Proposed Approach

In this section, the evaluation of the proposed methods using our simulations is described. The simulations are set up following the methodology explained in the previous section. The rooms designed for these simulations have medium sizes (i.e., no more than 6 × 6 m^2^—see Appendix A). As previously stated, we assume the robot is equipped with a 2D Lidar. However, for a better evaluation, within the same scenario, we use three Lidars placed at three different heights (0.03 m, 0.12 m, and 0.25 m, respectively). None of the Lidars interfere with the others, and they are just used for comparison. During the simulation of a single scenario, the simulated person performs different activities, seamlessly transitioning between them.

In the remainder of this section, we introduce the simulation parameters used and the data acquired. We then show the hyper-parameters of our neural networks, before proceeding with the evaluation of our proposed method.

### 6.1. Simulation Parameters

As previously described in Section 4, the Lidar simulator is meant to have a high flexibility and to be able to reproduce the behavior of a real-world Lidar. Thus, we introduced a certain number of parameters. The values used for these parameters are given in Table 1.

Apart from these parameters, the Lidar and environments themselves have their own parameters. These are given in Table 2.

Regarding our proposed approach, we have introduced a few parameters for our method given in Algorithm 1. These parameters’ values are given in Table 3.

Finally, we trained our neural networks for 500 epochs each, using a batch size equal to 16 and a learning rate equal to 0.001 with an Adam optimizer.

### 6.2. Simulation Results

#### 6.2.1. Output Visualization

For a better understanding, a visualization of the detection using a 2D Lidar is given in Figure 6. In the figure, we show an empty room with four static Lidars placed in the corners in addition to mobile Lidars. To recall, we equipped our robot with three Lidars at different heights for comparison. As can be seen from the figure, each of the Lidars detect the data from their own perspective. In addition, it is obvious from the figures in the blue area that the height of the Lidar affects the detection of the person when laying down on the ground (after a fall, for example); thus, the choice of the height of the Lidar is very crucial.

In Figure 7, we show a more realistic set of data collected in our actual scenarios. Here, the blue area shows the frames captured as seen from the Lidar’s perspective. In other words, in the center of the disks shown is the Lidar. The blue and red dots correspond to the detected data points. For these same frames, in the absolute reference, we show the data points detected by the mobile Lidar in their raw format (orange area), after transformation using the variations 1, 2, and 3 of the technique of transformation (green, gray, and red areas, respectively).

#### 6.2.2. Data Acquisition

Given the parameters set for our simulations, we created scenarios including all the activities. In our scenarios, we assumed a room of size 6 × 6 m with some obstacles. The data were performed continuously. Using a sliding window of frames, we created a data set of short samples of data, each lasting 2 s. The data set obtained is given in in Table 4. Being scarce, some activities are present in very few samples in the training and test sets. This is due to the fact that the original set of animations acquired does not have all the activities equally existing. To avoid problems of overfitting or data leakage, we did not use the same animation in the training and testing scenarios.

#### 6.2.3. Evaluation Metrics

In our work, the terms true positive (TP), true negative (TN), false positive (FP), and false negative (FN) for activity recognition are defined for a given activity as follows:True positive = the predicted label of the sample is the same as the label of the activity in question.False negative = the sample of the activity in question was wrongly given the label of another activity.True negative = a sample from a different activity was indeed given a label that is not that of the activity in question.False positive = a sample from a different activity was wrongly given the label of the activity in question.

Unsing these terms, we define our evaluation metrics as follows:Accuracy: The accuracy reflects how good the overall classification is. It shows how many samples are correctly classified compared to the total number of samples. It is expressed as follows:
(4)Accuracy=(TP+TN(TP+TN+FP+FN.Precision: The precision reflects how good the classifier is at identifying a certain class without confusing it with another. For the instances classified as belonging to that class, it computes the ratio of those that indeed belong to it. It is expressed as follows:
(5)Precision=(TP(TP+FP.Recall: The recall reflects how good the classifier is at classifying the instance of a given class. In other words, out of all the instances of a given class, it computer the ratio of those that were classified as belonging to it. It is expressed as follows:
(6)Recall=(TP(TP+FN.F1-Score: The F1-score is a metric that combines both the precision and recall, to address issues related to the misleading values of these two metrics when an unbalanced data set is use. It is defined as follows:
(7)$$F1-\mathrm{score}=2\;\cdot \;\frac{\mathrm{Precision}\;\cdot \;\mathrm{Recall}}{\mathrm{Precision}+\mathrm{Recall}}.$$

#### 6.2.4. Classification Results

In this subsection, we evaluate the performance of our proposed approach on the test set. As previously stated, we trained our neural networks for 500 epochs each, using a batch size equal to 16 and a learning rate equal to 0.001. The results reported are for the final epoch.

(a)Data representation techniques comparison

We first evaluate our approach using the three different variations of data transformation described in Section 5.2. Using these three variations, the validation set performance metrics are shown in Table 5. Despite being simpler and despite using a smaller image size, the Boolean grid variant performs better than the other two as validated in this table. The values for all the performance metrics are higher. This goes along with our initial intuition because there are many more features in this type of representation of the points than in the other two. While all the points are grouped in a single cluster in the circles and the ellipses cases, the Boolean grid variant groups the points in multiple clusters (one for each cell in the grid).

(b)Proposed method classification results

Since the Boolean grid technique has given the highest accuracy, we report its results in the rest of this work. We first report the performance results on test sets.

In Table 6 and Table 7, we show the performance metrics per class as well as the confusion matrix of the classification. As can be seen, the overall accuracy reaches 96.8%, whereas the individual activities’ accuracy ranges from 73.0% for the activity “Running” to 99.0% for the activity “Lying down”.

From the confusion matrix, we can see that some activities tend to be confused with one another. In particular, the class “Running” seems to be confused with the classes “Walking” and “Unsteady walk”. This is because these activities have similar patterns. From the robot’s perspective, the data points of the subject are moving over time. Since the window of time used is equal to 2 s, it is challenging for the LSTM network to correctly distinguish these activities. In addition, the direction in which the subject moves and that in which the robot moves affect the accuracy of the detection.

More importantly, we can notice that the activity “Falling down” has an accuracy equal to 81.2% and a precision equal to 88.5%. While this is low compared to other activities, it is important to notice that the event of falling occurs in a short period of time. In some instances, the person is blocked by some obstacles, making it impossible for the Lidar to detect them. That being said, it is still possible to identify such a fall given that the accuracy of the detection of the activity “Lying down” is equal to 99.0%. This activity naturally happens after the fall activity, and, even if the fall is not detected in real-time, the person can be identified later on as lying on the ground.

(c)Comparison with the conventional method [4]

Our work can be compared directly to the work [4], which uses a static Lidar placed in the environment. During our simulations, we placed four other Lidars in different positions in the room. This allows us to compare our method that uses the mobile robot to [4] under the exact same conditions.

In Table 8, we compare the performance of our proposed approach compared to that of [4]. Here, two of the four fixed Lidars’ results are reported: the one with the highest accuracy and the one with the lowest accuracy. As can be seen from the table, our proposed method by far outperforms [4]. The accuracy improvement is about 8% in the best case for [4] and higher than 16% for the worst case.

(d)Comparison with other existing method [8]

In addition to the work [4], we evaluated our proposed method to the one proposed in [8]. The approach was re-implemented in two different variations to fit with our data set format and specificities. In the first variation, we assume that a single Lidar fixed is used, whereas the second uses two Lidars located at opposite locations in the room. In this second variation, points of a single Lidar are taken into account at a time for a given sample. The Lidar taken into account is the one where the subject visibility is the highest. However, this could be misleading as the data format and type of scenarios we performed differ drastically. Therefore, we report for these methods their performance indicators in their data set as well. In Table 9, we show the performance of our proposed method against [8] with its different variations. Here, “Luo et al. [8]—original” refers to the performance metrics of the method [8] as reported in their original paper.

As can be seen, our proposed method, by far, outperforms the exiting method on our data set. This is to be expected given this method [8] has very strict constraints to perform well, namely the subject needs to be always in the Lidar’s field of view. The performance reported in [8] in the original work shows the potential of such approaches when such a requirement is satisfied.

### 6.3. Complexity Analysis

In addition to the evaluation metrics related to the performance of detection, if the proposed approach is to be implemented in real-world applications, the complexity of calculations and their being run on small, embedded devices needs to be addressed as well. Therefore, we evaluate our proposed method from this perspective. In summary, the algorithm of our proposed system is composed of three main steps:Step 1: the clustering of measurement points for every scan.Step 2: data transformation for every scan.Step 3: the classification of the entire sample.

We measure the individual complexities of these steps as follows:

#### 6.3.1. Complexity of Step 1

As seen in Algorithm 1, there are two nested loops iterating over the number of measurement points *N* in a scan, which present a complexity $O(N^{2})$, followed by the algorithm DBSCAN, which has a complexity $O(N^{2})$. Let us denote the number of rotations for a sample to classify $N_{r}$. The overall complexity of step 1 is then $O(N_{r}\times N^{2})$. We need to recall that the number of measurement points *N* is at most equal to 720 in our work and does not go to higher values. Similarly, the number of frames (i.e., scans) $N_{r}$ per sample is equal to 100 at most for 10 s worth of data.

#### 6.3.2. Complexity of Step 2

The way we transform the data relies on the grid size. Basically, for each data point in the cleaned-up scan (whose number of measurement points does not equal the number *N* defined in the previous step. Given a certain grid of size $X_{px}\times Y_{px}$ (referring to the number of pixels along the length and width axes), the pixel where one measurement point is located is given a value equal to 1, whereas a pixel where no measurement point is located is given a value equal to 0. Rather than iterating over the two axes’ pixels (which results in a complexity equal to $O(X_{px}\times Y_{px})$), we simply divide the coordinates of the measurement point by the length or width of 1 pixel, to directly derive its location. This reduces the complexity of this particular step (i.e., step 2) from $O(N^{2})$ to O(N).

#### 6.3.3. Complexity of Step 3

The classification complexity is measured in way that is different from the previous steps. The complexity of the neural network used to classify sequences of images is measured by evaluating the number of parameters, which reflects, in return, the number of operations performed. Given that the input of our neural network is a sequence of length $N_{t}$ of images of size $X_{px}\times Y_{px}$, we use the following formulas to evaluate the number of parameters. For a convolutional layer, the number of parameters is equal to the following:(8)$$P_{CONV}=\left(h_{input}\times w_{input}\times \left(n_{p_{filters}}+1\right)\right)\times n_{filters},$$
where $h_{input}$ and $w_{input}$ are the dimensions of the input images, $n_{p_{filters}}$ is the number of filters in the previous layer, and $n_{filters}$ is the number of filters of the layer in question. Note that we use a single layer in our case; thus, the value of $n_{p_{filters}}$ will be set to 1. However, we will keep the general formula.

For the LSTM layer, the number of parameters is equal to the following:(9)$$P_{LSTM}=\left(n_{input}\times n_{hidden}\right)+\left(n_{hidden}\times n_{output}\right)+n_{hidden}+n_{output},$$
where $n_{input}$ is the input size, $n_{hidden}$ is the size of hidden layer (number of neurons in the hidden layer), and $n_{output}$ is the output size (number of neurons in the output layer).

The number of parameters of a dense layer is equal to the following:(10)$$P_{DENSE}=n_{current}\times \left(n_{previous}+1\right),$$
where $n_{current}$ is the number of neurons in the current layer, and $n_{previous}$ is the number of values returned by the previous layer, be it another dense layer or a flattened output from a convolutional layer.

With all that in mind, our neural network architecture includes a time-distributed convolution (32 filters), one LSTM layer (with a hidden layer size equal to 40), and two dense layers (with 100 and 9 neurons at most).

Given a size of grid equal to 60×60 pixels, the total number of parameters is roughly equal to 18M. Note that the vast majority of the operations are when performing the computations in the LSTM layer. To put that in perspective, the total number of parameters of networks of image classifications, such as VGG16 and ResNet34, are equal to 138 M and 63.5 M, respectively.

#### 6.3.4. Overall Complexity

Given the complexities retrieved in the previous subsections, it is obvious that the classification step is the most consuming computation, the one that would present a bottleneck for the proposed system. However, there are a few things to keep in mind. For one, while the computation measured is performed for a given sample, in real-world implementation, the LSTM will process the frames sequentially, and return the output for every frame. In other words, for a new decision to be made, the computations performed to process all the scans (except the new one) will be reused. In addition, while computationally expensive for the embedded computer’s central processing unit (CPU), such computations are parallelized and can run much faster on its graphical processing unit (GPU).

### 6.4. Discussion

Given the nature of the task in hand, several challenges are yet to be addressed. While the proposed approach manages to identify the activity with an accuracy equal to 91.3%, the current work made a few assumptions that are not always valid. It assumes the subject and the robot to be in the same room all the time. Realistically, a cleaning robot in a multi-room living place means that the robot might be away from the subject for an extended period. In addition, multiple people could be present (e.g., an elderly couple), and our current study has not addressed the task of person identification based on the gait, nor do the models trained identify the activities of multiple people present simultaneously.

A more interesting point to address is the level of faithfulness of the simulations to the real world. A few assumptions were taken during the simulations. Even though we reproduced the behavior of the 2D Lidars in terms of inaccuracies and data loss, a few other aspects were not addressed. For instance, real-world Lidars do not work well in the presence of black objects, as the infrared light is not reflected; thus, the distance to these objects in not measured. In addition, the effect of multi-reflections is considered to be minimal, when in reality it could create several wrong points in the presence of highly reflective objects (e.g., mirrors, etc.).

Finally, the results and findings of this work are yet to be validated in a real-world robot. Through simulations, several aspects of the algorithm, processing, and classification could be tested in order to find the optimal values and representations. These are to be tested in real-world scenarios to prove their validity. While the acquisition of real-world data is very challenging, mainly for hazardous activities such as falling, models built through simulations could be evaluated on small real-world data sets for validation.

## 7. Conclusions

This paper aimed to develop a robust solution for human localization and human activity recognition (HAR) using a 2D light detection and ranging (Lidar) sensor on top of a mobile robot. Throughout this paper, we introduced our simulator, which we created and used to run our simulations and validate our proposed algorithms for human localization and activity recognition. Our approach uses the data collected by the Lidar mounted on top of the robot. The data captured by the Lidar are transformed, interpolated, and compared to a reference state of the surroundings. A customized algorithm is used to locate the subject in the room. A convolutional LSTM neural network is then trained to classify the processed measurements and identify if a fall event occurs or has occurred. Through simulations, we showed that such a system can achieve an accuracy equal to 81.2% in fall detection and 99.0% in the detection of lying bodies. Compared to a conventional method, which uses a static Lidar, the accuracy reaches for the same tasks 69.4% and 88.6%, respectively. This translates into an improvement of over 10% in the detection accuracy of both activities.

## Figures and Tables

**Figure 1 sensors-23-02534-f001:**
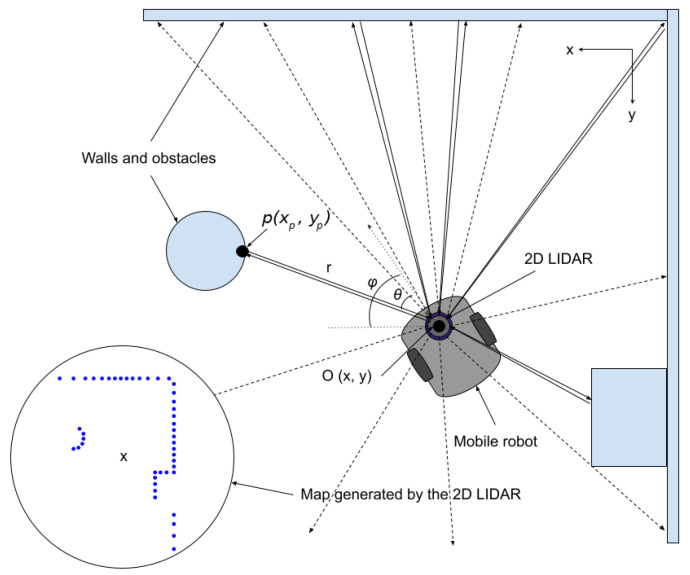
A simplified scheme of 2D Lidar’s functioning.

**Figure 2 sensors-23-02534-f002:**
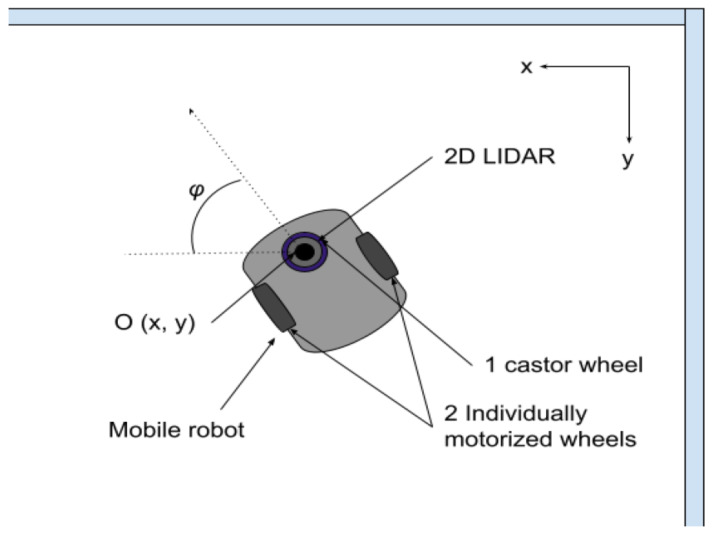
A simplified scheme of a differential drive mobile robot.

**Figure 3 sensors-23-02534-f003:**
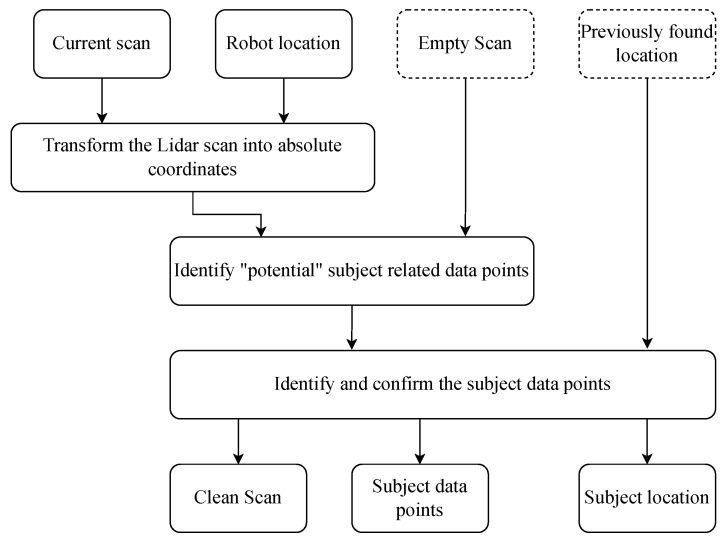
A simplified flowchart of the steps performed for location identification.

**Figure 4 sensors-23-02534-f004:**
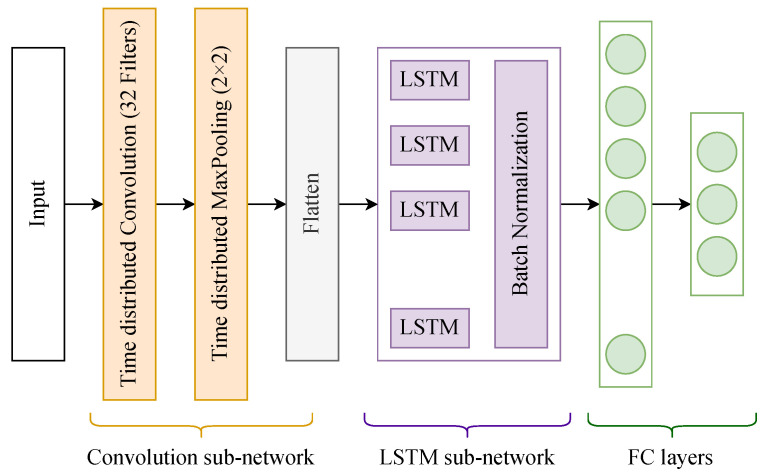
Architecture of the neural network used for classifying sequences generated by the variation 1 of our classification algorithm.

**Figure 5 sensors-23-02534-f005:**
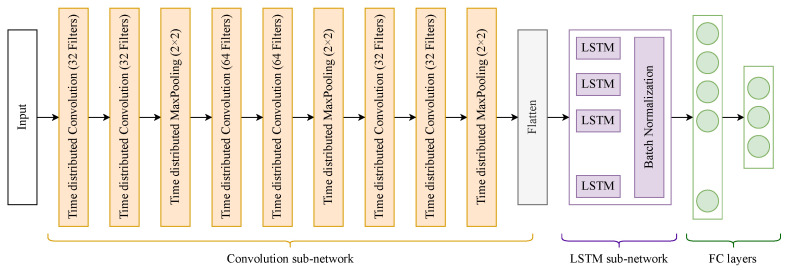
Architecture of the neural network used for classifying sequences generated by variations 2 and 3 of our classification algorithm.

**Figure 6 sensors-23-02534-f006:**
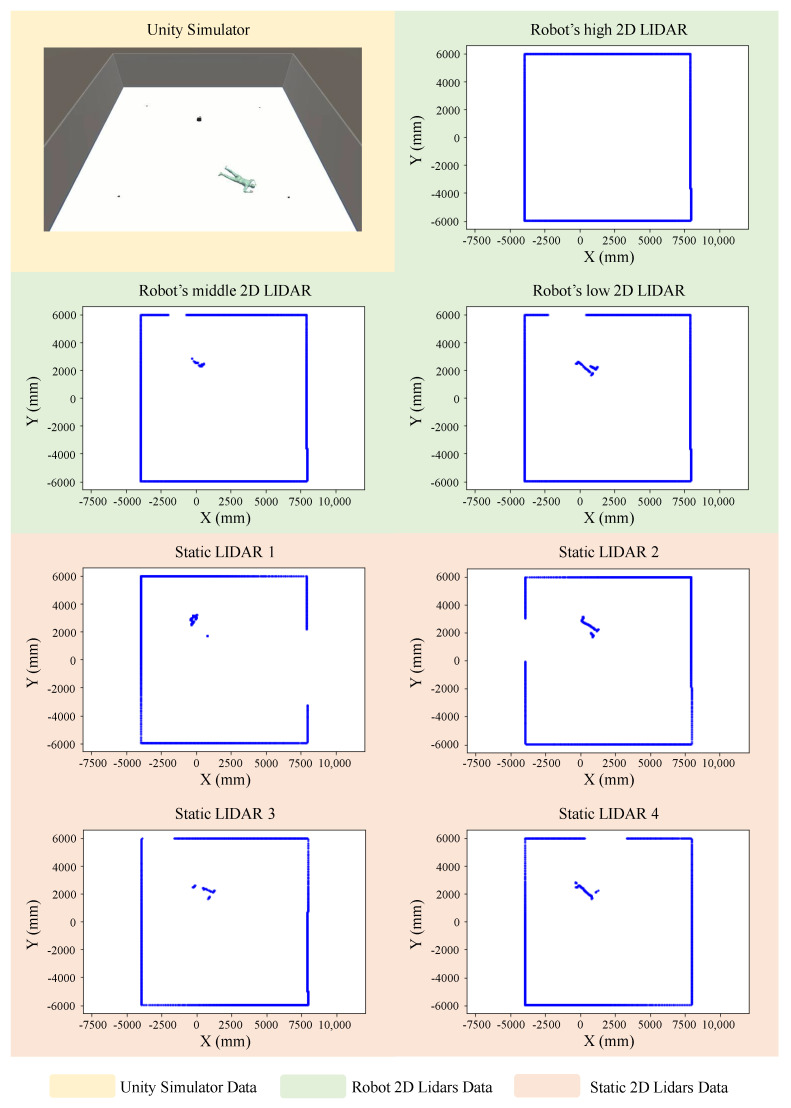
An example of an empty room with 4 static Lidars placed in the corners in addition to a mobile Lidars.

**Figure 7 sensors-23-02534-f007:**
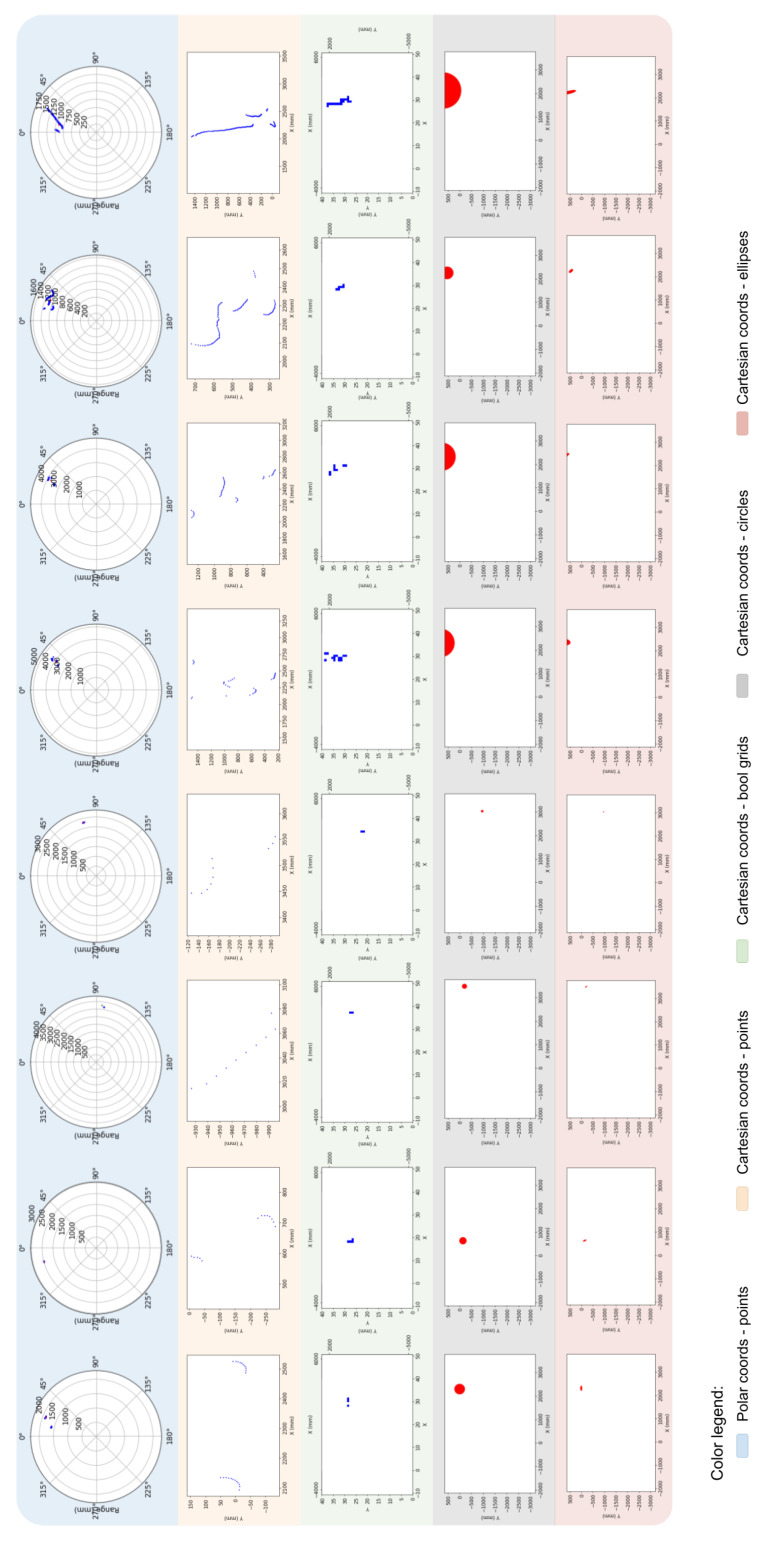
An example of data acquisition and transformation using our proposed technique.

**Table 1 sensors-23-02534-t001:** Parameters of the simulations meant to imitate real-world Lidars.

Parameter	Signification	Value
$d_{\mathrm{error}}$	The error in the distance measurement	2%
$N_{\mathrm{lost}}$	The percentage of measurement points lost	4%
$N_{\mathrm{extra}}$	The percentage of non-real points detected	2%
$\theta_{\mathrm{error}}$	The error in the estimation of the angle	0.04°

**Table 2 sensors-23-02534-t002:** Main Parameters of the Simulations.

Parameter	Signification	Value
Scan rate	The frequency of rotation of the Lidar	20 Hz
*N*	The number of measurements per rotation	720
$V_{\mathrm{robot}}$	The maximum velocity of the robot	0.2 m/s
$V_{\mathrm{human}}$	The maximum velocity of the human	1 m/s

**Table 3 sensors-23-02534-t003:** Parameters used for Algorithm 1.

Parameter	Signification	Value
$\delta_{d}$	The threshold to consider a data point as non-room-related	0.2 m%
$\Delta_{r}$	The min distance to a centroid for a point to be considered	0.8 m

**Table 4 sensors-23-02534-t004:** The data set acquired through our simulations.

Activity	Training	Test
Crawling forward	376	94
Falling down	340	85
Getting up	1460	365
Lying down	3608	902
Running	1112	278
Sitting on the floor	1580	395
Standing	3196	799
Walking	2948	737
Walking unsteadily	2760	690

**Table 5 sensors-23-02534-t005:** Classification results for the activity detection using the 3 variants of data representation.

	Boolean Grid	Circles Image	Ellipses Image
Accuracy	91.3%	78.8%	78.3%
Precision	91.3%	77.2%	76.4%
Recall	91.3%	78.8%	78.3%
F1-score	91.3%	76.5%	76.9%

**Table 6 sensors-23-02534-t006:** Classification results for the activity detection on the test set.

Activity	Accuracy	Precision	Recall	*F*1-Score
Crawling	88.3%	91.2%	88.3%	89.7%
Falling down	81.2%	88.5%	81.2%	84.7%
Getting up	93.4%	90.0%	93.4%	91.7%
Lying down	99.0%	99.3%	99.0%	99.2%
Running	73.0%	77.5%	73.0%	75.2%
Sitting	93.9%	95.1%	93.9%	94.5%
Standing	97.4%	95.3%	97.4%	96.3%
Walking	83.4%	87.0%	83.4%	85.2%
Unsteady walk	89.1%	85.1%	89.1%	87.0%
**Overall**	91.3%	91.3%	91.3%	91.3%

**Table 7 sensors-23-02534-t007:** Classification confusion matrix for the activity detection on the test set.

Activity	Classified as								
	(A1)	(A2)	(A3)	(A4)	(A5)	(A6)	(A7)	(A8)	(A9)
Crawling (A1)	**83**	2	5	2	0	0	0	0	2
Falling down (A2)	5	**69**	1	2	0	5	1	2	0
Getting up (A3)	1	3	**341**	1	0	5	5	1	8
Lying down (A4)	1	2	3	**893**	0	2	0	0	1
Running (A5)	0	0	2	0	**203**	1	1	49	22
Sitting (A6)	0	1	5	1	1	**371**	5	4	7
Standing (A7)	1	0	4	0	2	0	**778**	2	12
Walking (A8)	0	1	5	0	44	1	15	**615**	56
Unsteady walk (A9)	0	0	13	0	12	5	11	34	**615**

**Table 8 sensors-23-02534-t008:** Comparison of the classification metrics between our proposed method and that of [4].

	Proposed	Fixed Lidar (Highest)	Fixed Lidar (Lowest)
Accuracy	91.3%	82.9%	74.0%
Precision	91.3%	83.8%	75.5%
Recall	91.3%	82.9%	74.0%
F1-score	91.3%	83.4%	74.7%

**Table 9 sensors-23-02534-t009:** Comparison of the classification metrics between our proposed method and that of Luo et al. [8].

	Accuracy	Precision	Recall	*F*1-Score
Luo et al. [8]—variation 1	77.3%	78.4%	77.9%	77.9%
Luo et al. [8]—variation 2	82.7%	82.1%	82.7%	82.4%
Luo et al. [8]—original	99.4%	99.4%	99.5%	99.5%
Proposed method	91.3%	91.3%	91.3%	91.3%

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
