# Peer review of "A 2D-Lidar-Equipped Unmanned Robot-Based Approach for Indoor Human Activity Detection"

_sensors, 2023, doi:10.3390/s23052534_

Round 1
Reviewer 2 Report
This is an interesting topic, some comments are as follows:
(1)Please describe your contribution more clear, if just deploy the Lidar on the mobile robots, the novelty is not strong, based on this platform, please give the more description about the new problem and the contribution.
(2)The author mentioned that “To achieve such a goal, the measurements captured by the moving LIDAR are transformed, interpolated, and compared to a reference state of the surrounding.”How to transform, and how to interpolate, please give more details.
(3) In the simulation, does the parameter of the simulated person effect the data acquisition, please give some comments.
(4)The proposed method just use the simulation to verity the proposed method, the reviewer suggests that the author can carry out the field experiment to verity the proposed method if condition allows.
Round 2
Reviewer 2 Report
All issues I raised have been addressed, thank you for the authors, I think it can be accepted now.